

# Lots of movement, little progress: a review of reptile home range literature

Matthew Crane[1]   Inês Silva[2,3]   Benjamin M. Marshall[4]   Colin T. Strine[4]

[1] Conservation Ecology Program, King Mongkut's Institute of Technology Thonburi, Bangkok, Bangkhuntien / Bangkok, Thailand
[2] (CASUS), Center for Advanced Systems Understanding, Görlitz, Germany
[3] (HZDR), Helmholtz-Zentrum Dresden-Rossendorf, Dresden, Germany
[4] School of Biology, Suranaree University of Technology, Nakhon Ratchasima, Thailand

## ABSTRACT

Reptiles are the most species-rich terrestrial vertebrate group with a broad diversity of life history traits. Biotelemetry is an essential methodology for studying reptiles as it compensates for several limitations when studying their natural history. We evaluated trends in terrestrial reptile spatial ecology studies focusing upon quantifying home ranges for the past twenty years. We assessed 290 English-language reptile home range studies published from 2000–2019 via a structured literature review investigating publications' study location, taxonomic group, methodology, reporting, and analytical techniques. Substantial biases remain in both location and taxonomic groups in the literature, with nearly half of all studies (45%) originating from the USA. Snakes were most often studied, and crocodiles were least often studied, while testudines tended to have the greatest within study sample sizes. More than half of all studies lacked critical methodological details, limiting the number of studies for inclusion in future meta-analyses (55% of studies lacked information on individual tracking durations, and 51% lacked sufficient information on the number of times researchers recorded positions). Studies continue to rely on outdated methods to quantify space-use (including Minimum Convex Polygons and Kernel Density Estimators), often failing to report subtleties regarding decisions that have substantial impact on home range area estimates. Moving forward researchers can select a suite of appropriate analytical techniques tailored to their research question (dynamic Brownian Bridge Movement Models for within sample interpolation, and autocorrelated Kernel Density Estimators for beyond sample extrapolation). Only 1.4% of all evaluated studies linked to available and usable telemetry data, further hindering scientific consensus. We ultimately implore herpetologists to adopt transparent reporting practices and make liberal use of open data platforms to maximize progress in the field of reptile spatial ecology.

# INTRODUCTION

There are at least 11,242 described reptile species worldwide (*Uetz, Freed & Hošek, 2020*; accessed 2020-04). Terrestrial reptiles typically have narrower niche requirements and smaller ranges than other vertebrates such as birds and mammals, leaving them increasingly

Corresponding authors
Matthew Crane,
mattecology@gmail.com
Colin T. Strine,
strine.conservation@gmail.com

susceptible to threats such as habitat loss or invasive species (*Böhm et al., 2013*). Nearly one in five reptilian species are threatened with extinction (17.2%), and 14.4% are Data Deficient (*IUCN, 2021*; *Böhm et al., 2013*). Data deficiency in reptiles is higher than that of bird and mammal species (0.5% and 14.2%, respectively; *Stattersfield, Bennun & Jenkins, 2004*; *Schipper et al., 2008*); in particular for tropical reptiles and those with fossorial habits. Recent collapses of snake diversity have been reported with rippling effects to the ecosystem (*Zipkin et al., 2020*), but baseline data is often unavailable to properly evaluate these events and is likely understating the cascading effect of disappearing species (*Roll et al., 2017*). Many reptile species lack adequate baseline knowledge to inform conservation actions (*Tingley, Meiri & Chapple, 2016*; *Etard, Morrill & Newbold, 2020*). Spatial ecology datasets can help fill these baseline knowledge gaps: revealing how animals react to human changes to the landscape (*Tucker et al., 2018*), informing species conservation status (*Fraser et al., 2018*), and offering key prior information to design population assessments (*Gupta et al., 2019*).

Efforts to collect baseline data are hindered by reptiles' natural history—often small, rare, and cryptic—limiting detection probability. Telemetry studies can counteract low detection probability (or at least provide baseline detection estimates; *Boback et al., 2020*), as we know exactly the number, identity, and location of radio-marked individuals in the study site—with many potential applications (*Refsnider et al., 2011*). By tracking animal movement, we gain valuable insight into habitat requirements, foraging strategies, and behaviour (*Kingsbury & Robinson, 2016*).

Radio-telemetry (VHF) is common in terrestrial reptile research, whereas the use of GPS and other automated telemetry technology in terrestrial reptiles is still relatively rare (e.g., *Hart et al., 2015*; *Smith et al., 2018*) compared to other taxa (*Joo et al., 2020*). Using novel technologies and the resulting increased data volume in telemetry studies should also increase the analytical complexity and encourage greater uptake of more appropriate statistical methods. However, the uptake of modern techniques by practitioners within movement ecology has stagnated as the proportion of studies using movement-based area estimation methods is not matching the available software tools and methods (*Joo et al., 2020*).

The term "home range" is frequently and irrespectively applied to two distinct concepts: (1) the *Burt (1943)* home range definition, i.e., the area an animal uses for all of its lifetime activities, (2) within sample "space-use" (still commonly referred to as a "home range"), i.e., an area used by an animal throughout the study period duration. While both concepts have biological value, the chosen research question should govern choice of concept, and thus the space-use estimation methods researchers should use to answer their question. Researchers often use terms like "seasonal home range" to refer to animal space-use within the study period (*Viana et al., 2018*), delineating boundaries of interest based on season (*Korbelová et al., 2016*). Many studies improperly use the term home range (which by definition will include areas the animal used outside the study period, i.e., beyond-sample), when they intend to estimate short-term space-use of their animals (within-sample). Home ranges (in either of its definitions) and movement pattern data can help us understand population dynamics and habitat use, informing protected area

size or policy processes (*Metcalfe et al., 2015*; *Fraser et al., 2018*), advocating for specific land-tenure systems (*Johansson et al. 2016*; *Farhadinia et al., 2018*), and recovery planning for threatened species (*Parsons, 2016*).

We treat reptile home range studies as any study intending to quantify space-use, regardless of whether the intent was to estimate areas used outside of the study or bounded by the study period. Initially, geometric methods such as the Minimum Convex Polygon (MCP) were the norm, but subsequently researchers have turned towards statistical techniques incorporating underlying probabilistic models, such as Kernel Density Estimators (KDE; *Worton, 1989*). The autocorrelated nature of movement data violates traditional KDEs assumptions led to the development of movement-based area estimation methods: autocorrelated KDEs (AKDEs; *Fleming et al., 2015*), and Brownian Bridge Movement Models (*Horne et al., 2007*; *Kranstauber et al., 2012*). Although researchers have continued to expand and develop analytical methods within movement and spatial ecology (*Laver & Kelly, 2008*), the proportion of studies using movement-specific methods has not increased (*Joo et al., 2020*).

*Macartney, Gregory & Larsen (1988)* summarized the landscape of snake home range studies and suggested developing useful baseline data for comparative purposes requires longer-term studies and standardised data collection, analysis, and presentation. In 1990, a general review found most studies focused on mammals and used MCPs to estimate home ranges (*Harris et al., 1990*). By 2008, the same patterns were still present, with mammalian and ornithological home range studies most prevalent, and with 96 out of 141 studies still utilizing MCPs (51% utilizing both MCPs and KDEs; *Laver & Kelly, 2008*). *Goldingay (2015)* reviewed home-range studies for Australian terrestrial vertebrates between 2001–2012, and only 19% out of 150 papers pertained to reptiles, even though Australia has over 860 native reptile species (39% of Australian's land species). As for home range estimators, MCPs appeared in 84% of these studies, followed by KDEs (45%), illustrating a lack of methodology advancement despite a growing field. Taken together, previous reviews suggest the potential for shortfalls in reptile spatial studies and reliance on MCP or KDE methods. If trends in data missingness also apply (*Etard, Morrill & Newbold, 2020*), the shortfall in studies may be greatest in the tropics where reptiles are most diverse (*Roll et al., 2017*).

Here, we reviewed reptile telemetry literature to assess whether the field has shifted collection methods (e.g., GPS and satellite tags), and participated in the uptake of newer home range estimation techniques. We also sought to reveal underlying reptile home range study biases, both geographically and taxonomically, to determine future limitations in undertaking global syntheses and analyses. As most home range estimates are sensitive to study design and data collection protocol (e.g., number of locations and duration), we also evaluated reptile telemetry studies within the framework of open, reproducible, and comparable science, to determine the number of available datasets from our review. Finally, we make recommendations for improving reporting standards to aid in making reptile home range studies more broadly applicable and reproducible.

## Survey methodology

We performed a comprehensive literature review by searching in Google Scholar, Web of Science, and Scopus on 30th of January, 2020 for articles relating to reptile spatial ecology using the terms ("reptile" OR "tortoise" OR "crocodile" OR "alligator" OR "snake" OR "lizard") AND ("home range" OR "home-range" OR "space use" OR "spatial ecology"). Following *Haddaway et al. (2015)* we only included the first 300 results from Google Scholar. We limited the search to papers from 2000-2019 published in peer-reviewed journals, because we were interested in the changes/uptake of different computational home range methods (prior to 2000 researchers had to rely heavily on manual cartographic methods).

Our aim was solely terrestrial/semi-terrestrial reptile home range studies, so we excluded studies on marine species (e.g., sea turtles, sea snakes). However, we did include studies from semi-aquatic or typically range-limited to waterway species (e.g., crocodilians, freshwater turtles). We did not include marine species as they represent unique challenges and opportunities for modeling space-use such as 3D space use. We excluded studies lacking home range or space-use area estimates, such as those that only used movement measurements. As multiple field sampling techniques can generate home ranges, we defined our inclusion criteria as only studies using an attached telemetric device (e.g., VHF transmitters, GPS). We further excluded clear re-analyses of previously published datasets to avoid pseudoreplication. In these cases, we included only the oldest published article returned from the systematic search for review. However, we did include studies pooling previous data with newly collected data.

When studies included multiple species, we considered the overall methodology rather than for each species individually to avoid pseudoreplication. We only collected multiple values for a study's methodology if researchers used two different tracking devices (e.g., both VHF and GPS), as different tracking devices are subject to different limitations in data recording. We used the distinct biotransmitter type to review tracking protocols (e.g., sample frequency, number of locations) and trends in biotransmitter selection (e.g., VHF vs GPS), but used study level effort to review geographic and taxonomic patterns. From each included paper, we collected basic study information (country, year, species, number of individuals tracked) as well as more detailed information about the data sampling regime and home range estimation methods. To assess the field sampling protocols, we collected data concerning the reporting of tracking duration, number of locations, and tracking frequency (number of fixes per day) for studied individuals.

Regular temporal sampling is an assumption in several movement analyses, so we also identified whether studies conducted regular sampling. We defined two cases of regular temporal sampling: (1) sampling occurred at an equal hourly sampling rate (e.g., one fix every 2 h, one fix every 15 min), (2) individuals were located at least once per day consistently throughout the study (e.g., data with temporal resolution sufficient for subsetting to one location for every day the animal was tracked). We converted the reported tracking methodology into the number of tracks per day, recording both the minimum possible and maximum possible frequency. In cases where authors used ambiguous language (e.g., biweekly), or provided insufficient detail, we classified the tracking frequency as "not
**Table 1  Scoring category definitions for both number of location and study durations.**

| Data Field Type | Score of 0 | Score of 1 | Score of 2 | Score of 3 |
|---|---|---|---|---|
| Duration Reproducibility | *No reporting* - includes cases when authors report at least ## days without a maximum | *Population-level reporting -* mean only *or* sum of duration (as number of days or number of weeks) | *Population-level reporting* - Mean only *or* sum of duration + a metric of spread (such as standard deviation or error) | *Individual-level reporting -* actual date ranges or number of days for each individual included in the paper. |
| Location Reproducibility | *No reporting* - includes cases when authors report at least ## locations without a maximum | *Population-level reporting -* mean only *or* sum of number of locations. | *Population-level reporting -* Mean only *or* sum of number of locations + a metric of spread (such as standard deviation or error) | *Individual-level reporting* - actual number of times each individual was located during the study. |

reported". We also documented whether the study used multiple regular tracking regimes (e.g., tracking once per day in the summer months, and only weekly during the winter).

We coded each article's adherence to two key reporting characteristics that can impact space-use and home range estimates: tracking time duration, and number of fixes. Tracking time duration differs from study duration and refers to the period of time over which researchers tracked an individual. In contrast, study duration is the overall study period, and thus represents a study-level characteristic, while tracking duration represents an individual-level characteristic. We scored articles on a scale of zero to three. For example, zero indicated reporting only study duration/study-level number of fixes (e.g., "...tracked individuals from 2018-01-01 to 2018-09-23..." or "...collected a total of 356 fixes...") while failing to report the exact data quantity per individual (See Table 1 for details).

For each included study, we also recorded the method for estimating home range area. Kernel Density Estimation is a common technique but is highly dependent on the smoothing factor (h) selection method. To address this, we recorded the method used to determine the h-value for KDEs (when reported). We recorded whether the authors reported a movement metric based on time (e.g., mean daily displacement), as field sampling regime can also affect such metrics. Finally, we recorded if the study attempted to "validate" the home range estimation—i.e., included any form of analysis that assessed the relationship between number of locations and the home range area estimate (e.g., linear regressions, bootstrapped asymptotes). Each source was assigned a primary reviewer from the author team; however, any ambiguities in how a source should be coded was flagged and reviewed by all authors to remain consistent.

## Exploratory model

We used a Logistic Bayesian Regression Model to explore the relationship between a reptile's body mass and the likelihood of them being studied. We used the log10 body mass data compiled by *Meiri et al. (2021)* and matched it to the study species within our literature search. We manually corrected 25 names that had changed or had been misspelt in our dataset. We inserted missing body mass data for *Platysternon megacephalum* with results from *Sung, Hau & Karraker (2014)* suggesting an average body mass of 393.3 g.

We used the resulting log10 body mass values to predict a binary of whether a species was studied (Bernoulli distribution), and allowed both gradient and intercept to vary based on

order: studied $\sim 1 + \log 10$mass $+ (1 + \text{order} \mid \text{order})$. We excluded orders Rhynchocephalia and Squamata (Amphisbaenia) because of the low species richness and zero studied species respectively, and (sub-)families Hydrophiinae, Dermochelyidae, and Cheloniidae as they are predominantly marine (final model $n = 11,037$ species). We implemented weakly informative priors cauchy(location = 0.1, scale = 3) for the log10mass coefficient and cauchy(location = 0, scale = 1) for the standard deviation between orders.

We ran the model using four MCMC chains each with 5000 iterations and 2000 warmup, then thinned by a factor of two. To achieve convergence, we increased the adaptive delta to 0.999 and maximum tree depth of 15. We assessed model convergence using $\hat{r}$ values $\sim 1$ and visual examinations of autocorrelation and trace plots.

### Data and software availability

We used R v.3.6.3 (*R Core Team, 2020*) and RStudio v.1.4.1029 (*RStudio Team, 2020*) to summarise all data. We summarised data with dplyr v.1.0.2 (*Wickham et al., 2020*), raster v.3.4.5 (*Hijmans, 2020*), forcats v.0.5.0 (*Wickham, 2019a*), reshape2 v.1.4.4 (*Wickham, 2007*), stringr v.1.4.0 (*Wickham, 2019b*), and tidybayes v.2.3.1 (*Kay, 2020*) packages. We ran Bayesian Regression Models using brms v.2.14.2 (*Bürkner, 2017*; *Bürkner, 2018*) and rstan v.2.21.2 (*Stan Development Team, 2020*). We visualised data with cowplot v.1.1.0 (*Wilke, 2019*), ggplot2 v.3.3.2 (*Wickham, 2016*), ggpubr v.0.4.0 (*Kassambara, 2018*), ggrepel v.0.8.2 (*Slowikowski, 2018*), ggridges v.0.5.2 (*Wilke, 2018*), ggtext v.0.1.1 (Wilke, 2020), and scico v.1.2.0 (*Pedersen & Crameri, 2018*).

We have included all data, summary code, and model output at Zenodo: (http://doi.org/10.5281/zenodo.4303643). Data file "reptileHRReview_References.csv" includes the results of the stages of the literature review alongside article information; the data file "reptileHRReview_LiteratureReview.csv" contains the raw data from the literature review process; metadata file "reptileHRReview_Metadata.csv" includes full descriptions of all columns in both main data files. An additional file from Reptile Database (*Uetz, Freed & Hošek, 2020*; accessed 2020-04) "reptileChecklist_2020_04.csv" includes the information used for genus- and clade-based summaries. We created the reptile diversity using Global Assessment of Reptile Distributions (GARD) data (representing 99% of reptile species' distributions at time of its publication [10,064 species]; *Roll et al., 2017*) and functionality from sf v.0.9.6 (*Pebesma, 2018*) and fasterize v.1.0.3 (*Ross, 2020*) packages. We counted terms used and produced the word cloud using pdftools v.2.3.1 (*Ooms, 2019*) and quanteda v.2.1.2 (*Benoit et al., 2018*).

## RESULTS

### Data collection

From 1,028 unique articles returned from the literature searches (Fig. S1), our exclusion criteria produced a final sample of 290 reptile spatial ecology studies consisting of 302 tracking subsets (accounting for multiple tracking protocols, i.e., GPS and VHF, within each study) involving 7,861 individual animals. However, one study failed to report the number of animals tracked. The majority of studies used VHF telemetry devices (277), with 22 using GPS, and a further three instances of ultrasonic or satellite tracking.

Regardless of the tracking method used, tracking frequency varied dramatically (Fig. 1): ranging from 480 (24 if automated VHF is excluded) to 0.0328 tracks per day for VHF, and 144 to 0.143 per day for GPS. In other words, tracking 0.0328 times per day is equivalent to tracking approximately once a month (i.e., 1/30.5), and 0.143 times per day is the equivalent to tracking once per week (i.e., 1/7). Eighty-eight tracking subsets (29.4%) had consistent tracking frequencies throughout the study (i.e., minimum and maximum tracking frequency are the same, with no seasonal variation or multiple tracking regimes), and 65 tracking subsets had a tracking frequency of at least once per day (i.e., minimum tracking frequency greater than one throughout the study period). The number of tracks/fixes per day was not always reported ($n = 26$; 9%), or reported ambiguous maximum and minimum number of tracks per day (e.g., "bi-weekly", "at least"); this number increases to 78 tracking subsets that failed to clearly report one extreme of the tracking frequency. Such reporting is key when measurements of movement rate (or any metric that incorporates time) are calculated, and 190 out of 290 studies reported a movement metric.

Despite the extensive field effort expended tracking 7,861 animals, we identified serious gaps in basic reporting that undermine understanding basic study characteristics. The gaps in reporting appear relatively consistent over the 20 years reviewed (Fig. 2). In addition to the 26 instances of incomplete tracking frequency data (Fig. 2A), 162 (56%) studies provided very limited or missing descriptions of tracking duration (135 [47%] scored 0, 27 [9%] scored 1; Fig. 2B), and number of fixes obtained (95 [33%] scored 0, 52 [18%] scored 1, sum 147 [51%]; Fig. 2B). Reporting standards of 2 and higher (i.e., likely sufficient to enable meta-analyses inclusion) were reached in 128 [44%] studies for durations and 143 [49%] for the number of fixes. Location reporting was further hindered by ambiguous terms, we found 34 different terms describing how many times an animal was tracked: studies largely used terms stemming from locat*, but even within a single study we often found multiple terms used to describe when researchers located animals (Fig. S2). Providing raw data could mitigate reporting deficiencies; however, we found only 24 [8%] studies included links to external data and only 4 [1% of all studies] of those links led to raw tracking data.

## Estimation methods

Between 2000–2019 the number of studies per year increased from 6 in 2000 to 18 in 2019, with a low of 4 in 2001 and a peak of 25 in 2017 (Fig. 3A). Minimum Convex Polygons (MCP) and Kernel Density Estimations (KDE) use has dominated reptile home range studies for the past 20 years (272/290 studies; Fig. 3B) and were present in over 75% of studies each year (Fig. 3B). Frequently, studies include estimations from both methods, and rarely use KDEs without including MCPs (Fig. S3). A minority of studies ($n = 19$) used "other" methods without pairing to estimations via MCPs and KDEs. These methods included: alpha-hull methods, harmonic means, linear home ranges, Brownian Bridge kernels among others (for full list see Table S1). Of all other methods listed, only dynamic (and standard) Brownian Bridge Movement Models directly incorporate movement to estimate space-use (i.e., movement models).

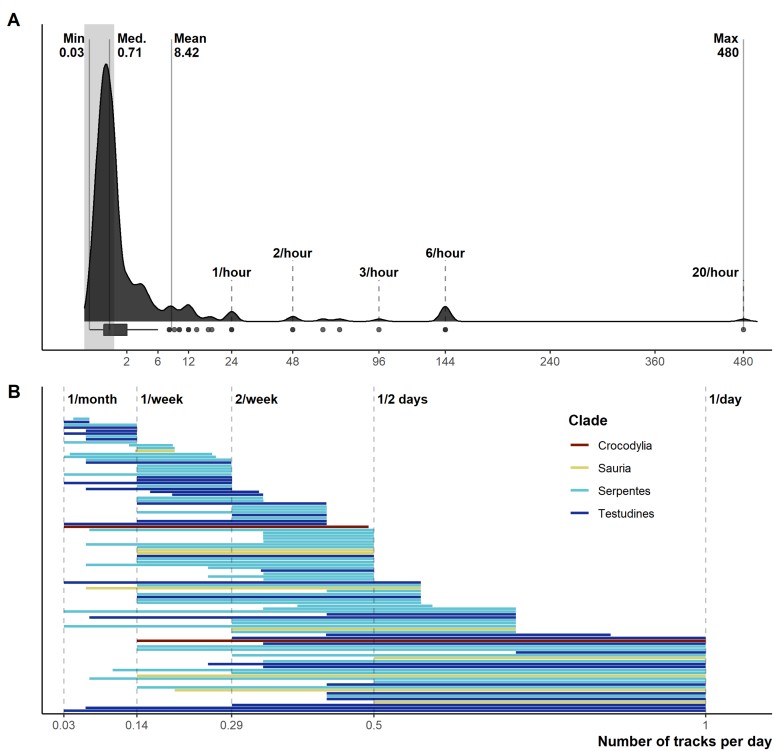

**Figure 1** **Reported number of tracks per day.** (A) Density and boxplot plot showing the distribution of all reported tracking frequencies (minimums and maximums), $x$-axis is square-root transformed. Shaded grey area highlights instances of < 1 track per day. Text labels show the minimum, median, mean, and max numbers of tracks per day. (B) The variation per study between the maximum and minimum tracking frequencies, provided both are < 1 per day. Both plots exclude all unreported tracking frequencies.

Studies using MCPs largely made use of high % contours (100% and 95%; Fig. S4). KDEs used a greater diversity of contour values (5 to 100%), but with clear concentration towards 95% and 50%. Studies more frequently (n = 97/270 studies using MCPs) failed to report the contour used with MCPs than other methods, potentially connected to the assumption that MCPs default to 100%.

For studies using KDEs, we found 14 smoothing factor selection methods, but researchers primarily used Least Squares Cross-validation (LSCV; 73/159; Fig. S5). Similar to basic reporting, we show that 27 (17.4%) studies failed to report a smoothing factor, either by omission or by only stating the "default" for a software.

## Geographic and taxonomic biases

The United States of America is a clear hotspot where 133 of 290 studies were conducted. All other countries are dramatically lower (<8 studies, 30 countries with a single study), with only Australia (35), Canada (22), and South Africa (12) breaking the trend. Despite high reptile diversity, Africa exhibited a dearth of reptile home range papers (Fig. 4).

The 7,861 tracked individuals, 302 tracking subsets, and tracking subset sample sizes were not split evenly across the major clades of Crocodylia (mean individuals per subset = $11.1 \pm 1.62$), Serpentes ($23.6 \pm 1.71$), Sauria ($28.4 \pm 3.53$), and Testudines ($35.4 \pm 6.80$;

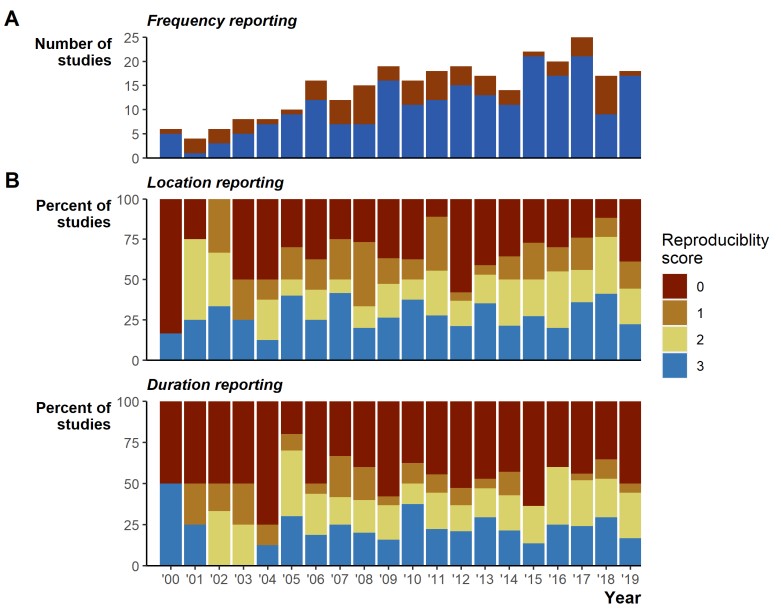

**Figure 2** **Reporting scores over time.** (A) Whether the study fully reported tracking frequency (complete in blue versus incomplete in red). (B) Percent of studies scoring 0 through 3 on location and duration reporting, where a score of 3 is the most complete reporting.

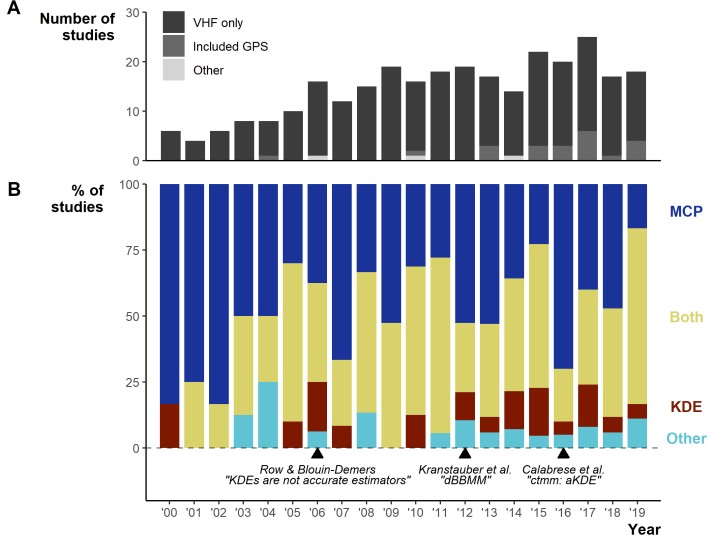

**Figure 3** **Changes in the field from 2000 to 2019.** (A) Number of articles over per year and the telemetry devices used: dark grey = only VHF used, middle grey = GPS was used exclusively or in conjunction with VHF, light grey = other device used (ultrasonic and satellite). (B) The percentage of studies using Minimum Convex Polygon (MCP), Kernel Density Estimations (KDE), both or other estimation methods. "Other" only includes studies that did not use either MCPs or KDEs. Lower text labels highlight the year select papers were published aiming to guide, or enable new, space-use estimation.

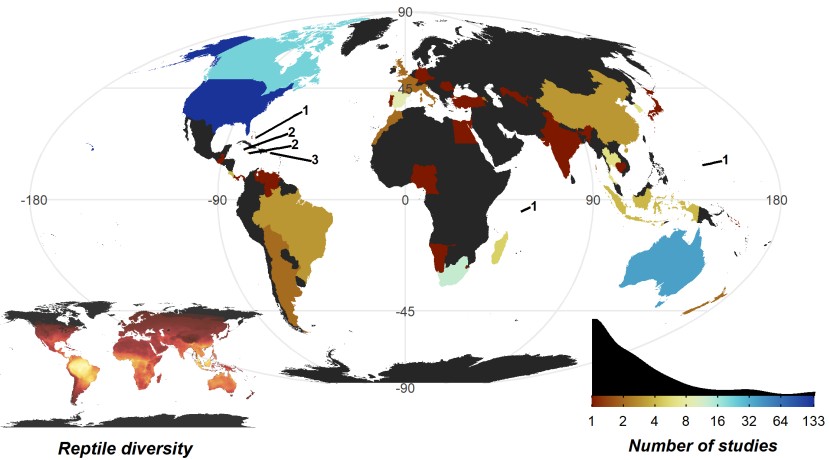

*Reptile diversity* | *Number of studies*

**Figure 4  Number of studies undertaken in each country.** Colour scale is detailed in the insert density plot bottom right [blues indicating more studies, reds indicating fewer studies, grey indicating zero studies], showing the distribution of per country study counts. Count of studies is shown on a log scale to help differentiate between countries with fewer studies, a diverging colour scheme was selected to highlight the high-count outliers. Smaller territories are highlighted with a label denoting the number of studies. Insert map bottom left, shows the distribution of reptile species globally, ranging from zero species (black) to 182 species (yellow); the heatmap was generated using GARD data (*Roll et al., 2017*).

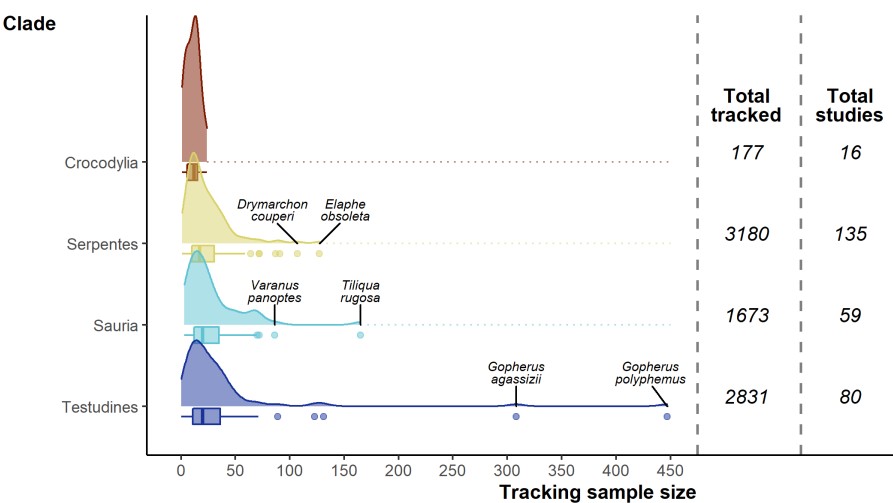

**Figure 5   Density and box plots showing the distribution of sample sizes (tracked individuals) per study by clade.** Species names highlight the top two outlying sample sizes for clades other than Crocodylia.

Fig. 5). Serpentes was the most studied clade and with the most tracked individuals, whereas Crocodylia was the lowest. However, in terms of percentage of genera studied, Crocodylia leads with 44.4% (4/9; Testudines 27/94, 28.7%; Serpentes 40/522, 7.66%; Sauria 28/564, 4.96%).

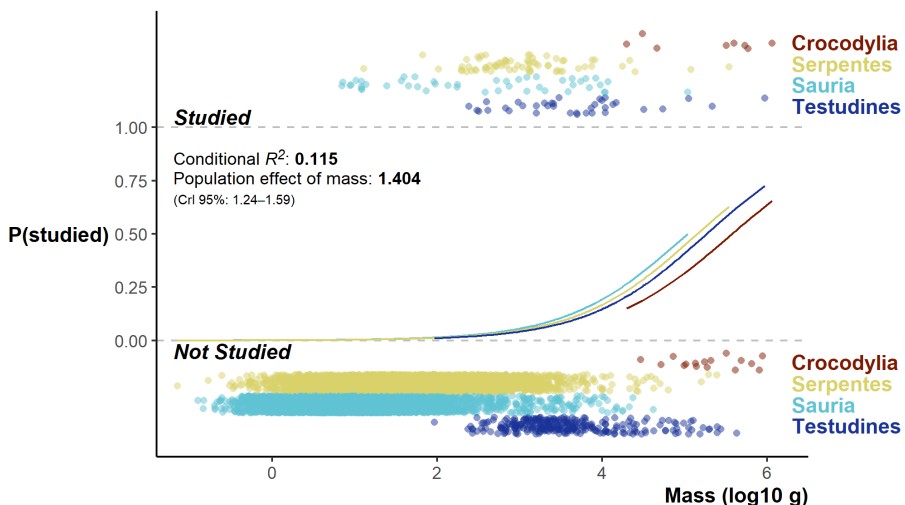

**Figure 6** Fitted model values split by group effect of order, flanked top and below by the distribution of studied and unstudied species log$^{10}$ body mass (jittered on the *y*-axis).

Overall, of the 1,186 terrestrial reptile genera (*Uetz, Freed & Hošek, 2020*; accessed 2020-04), 99 (8%) have been tracked (but there are genera untrackable with current telemetry equipment, e.g., Amphisbaena and Ramphotyphlops). Two genera (Crotalus & Gopherus) stand out having been studied 22 and 23 times (Fig. S6), whereas 45 genera had only a single study.

The Bayesian Regression Model successfully converged, revealing a very low base chance of a species being studied (and detected in the literature review): 0.06% (CrI 0.02 –0.18%). A species study chance increased at a population level with increased log10 body mass ($\beta_{\text{log10mass}}$ = 1.40, 95% median HDI CrI 1.23–1.58; Fig. 6); however, the overall model fit was poor (conditional R$^2$ 0.115, and marginal R$^2$ 0.113). While there were differences among the orders, the direction and magnitude of the differences was neither unambiguous nor large (Credible intervals all overlapped zero).

## DISCUSSION

We identified key issues limiting study comparability. Study design decisions regarding tracking frequency and duration are critical considerations when attempting to produce biologically relevant space-use estimates (*Girard et al., 2002*; *Börger et al., 2006*; *Silva et al., 2020*). These decisions determine total individual sampling effort; an individual with 12 locations over a single day is unequal to one with 12 locations over an entire year. Reporting solely the study duration (e.g., stating that tracking occurred between X and Y date) would then obscure individual variation—further hindering our ability to generalise across the study population. By reporting at the individual level, researchers can highlight potential sources of heterogeneity between studies (e.g., tracking an animal once every week while tracking others twice a week would hinder daily movement comparisons) (*Alexander & Maritz, 2015*; *Riotte-Lambert & Matthiopoulos, 2019*). As we found a wide range of tracking

frequencies throughout reptile spatial ecology studies (and gaps in reporting), it limits our ability to conduct large inter-study comparisons and undermines inter-study comparison validity.

Our review also reveals major biases in the study of reptile home ranges. Geographically, nearly 50% of studies originated from a single country, the United States of America. We found a stark mismatch between reptile diversity and reptile home range study locations (*Roll et al., 2017*), reflecting similar gaps seen in reptile abundance studies (*Doherty et al., 2020*); in particular, the Middle East and Central Africa. Taxonomically, we observed less severe biases, but should still be considered in evaluating the patterns in the available data. Only 8% of genera have been studied and a relatively small number of genera dominate the available reptile spatial ecology data (e.g., *Gopherus, Crotalus, Pituophis*). Although the model fit was poor, our results show a greater chance that larger (by log10 mass) species are studied. The more frequent tracking of larger species likely stems from limitations surrounding biotelemetry device size to body mass ratio, and attachment/implantation methods required to track species with fossorial and arboreal habits. Efforts to synthesise reptile home range or movement must recognise that any results may be biased towards patterns in larger temperate western hemisphere species, rather than global trends. Global syntheses may be inhibited by the drastic differences in seasonal climate between well-studied temperate areas and neglected tropical regions, which is likely key to reptiles as ectotherms (*Shine & Madsen, 1996*). If the exceptional value and extent of reptile diversity in tropical areas (*De Miranda, 2017*; *Roll et al., 2017*) is underappreciated due to lack of data or representation, global conservation strategies may inadvertently tailor to larger temperate western hemisphere species.

Many of the issues that we revealed in the reptile spatial ecology literature can be mitigated with greater transparency, adopting open science and reproducible analyses (i.e., code-based analysis avoiding language ambiguities by comprising, and describing, the exact analytical procedure performed; (*Ince, Hatton & Graham-Cumming, 2012*; *Archmiller et al., 2020*). Open science presents a vital resource for replication efforts and can facilitate better meta-analyses. It also benefits the original researchers by increasing citations, boosting publication chances, and creating more potential collaborations (*Piwowar & Vision, 2013*; *Markowetz, 2015*; *Allen & Mehler, 2019*). The disparity between reptile data and other taxa on MoveBank (a prominent movement data repository) re-emphasizes our review findings. When searching either ''reptilia'' or ''reptile'', 24 reptile studies have available movement data on MoveBank as of 2020-02-12, and only 11 of those studies focus on terrestrial species out of over 6,000 available studies. Ecology journals (and herpetology journals especially; *Marshall & Strine, 2021*) should redouble efforts to enforce data availability statements (*Roche et al., 2015*) and counteract the reluctance to biotelemetry data sharing expressed by late career researchers (*Campbell, Micheli-Campbell & Udyawer, 2019*), making data availability the default and refusing to accept ''on request'' statements (*Aalbersberg et al., 2018*). K'[ Researchers can make use of free data repositories (movement specific like MoveBank, or generic like Zenodo or OSF) to ease this process. We hope the opening of reptile movement data can facilitate broader studies similar to those undertaken in avian and mammalian fields (e.g., *Tucker et al., 2019*; *Noonan et al., 2020*).

Researchers often justify using KDEs and/or MCPs to compare with the wider reptile spatial ecology literature. However, methodological choices in reptile space-use studies hinder inter-study comparisons, as KDEs and MCPs are sensitive to differences in sampling effort (e.g., number of locations, tracking duration and frequency) (*Mitchell, White & Arnold, 2019*; *Silva et al., 2020*). Reptile studies also used a wide range of smoothing factors for KDEs, which can also result in considerable home range over- or underestimations (*Bauder et al., 2015*; *Silva et al., 2020*). For example, two widely used smoothing factors, href and LSCV, produce dramatically different area estimations. Failure to report or account for smoothing factors is thus a major concern, as it would significantly alter meta-analysis patterns. Although *Row & Blouin-Demers (2006)* suggested MCPs over KDEs for home range size comparisons across groups or time periods, MCP and KDE comparability is unreliable rendering their use generally inappropriate for most ecological studies (*Nilsen, Pedersen & Linnell, 2008*; *Silva et al., 2020*).

There is a growing body of work demonstrating the versatility of newer analytical methods (*Noonan et al., 2019*; *Silva et al., 2020*; *Silva et al., 2021*), and how they can be applied to the coarser resolution radio-telemetry data and the particulars of reptile movement (e.g., zero-inflated step lengths arising from long and frequent periods when the animal is stationary; *Averill-Murray, Fleming & Riedle, 2020*; *Hromada et al., 2020*; *Silva et al., 2020*). Reptile spatial ecology so far has largely failed to capitalise on the wealth of analytical options available, namely integrating movement information explicitly into estimations of space-use. Unlike traditional estimation methods (KDEs and MCPs), movement-based area estimation models do not operate under the assumptions breached by tracking data (independence of points) and guard better against under- and overestimation (*Fleming & Calabrese, 2017*; *Silva et al., 2020*). One of the common solutions to autocorrelation is the thinning of data; this procedure is inherently wasteful and inefficient, defeating the purpose of collecting high temporal-resolution data and reducing the biological relevance of telemetry datasets (*Fleming et al., 2015*; *Calabrese et al., 2021*). With low temporal-resolution data (typical of VHF data), analytic approaches will not necessarily reveal the correct home range patterns and need to be applied with caution; in these cases, it may be necessary to reconsider the research questions or re-evaluate study design for additional data collection. However, low sample sizes do not immediately exclude the use of newer methods; although typically designed to handle high volumes of data, AKDE will account for low sample sizes and can be used in conjugation with VHF data to obtain home range area estimates (*Fleming et al., 2019*). Similarly, dBBMMs can estimate movement pathways with low-volume VHF reptile data (*Silva et al., 2020*).

Conceptualising home range as within sample versus beyond sample space-use requires distinguishing between occurrence distribution versus range distribution methods (*Fleming et al., 2015*; *Horne et al., 2019*; Fig. 7). While occurrence distributions (e.g., dBBMMs) allow us to answer research questions regarding the movement trajectory of an animal (and its confidence region), range distributions (e.g., AKDEs) consider the processes underlying animal movements and long-term space-use (*Horne et al., 2019*). Some research questions investigated in the reptile home range literature are actually targeting within sample space-use, requiring no extrapolation beyond the sampling period. In many cases, the sampling

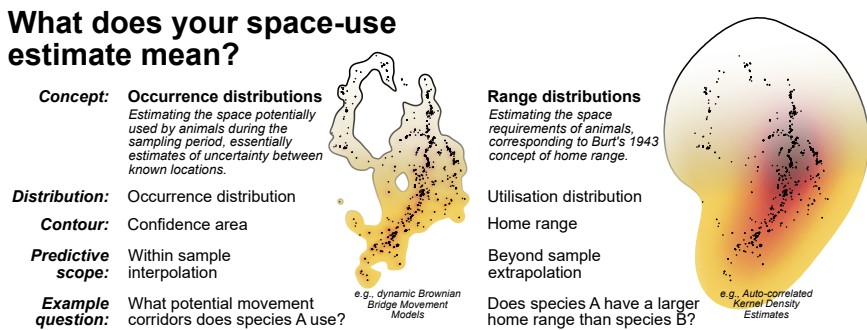

**Figure 7** **A breakdown of the two complimentary conceptualisations of animal space-use.** Displayed alongside are examples of the 99% contour derived from dynamic Brownian Bridge Movement Models (*Kranstauber, Smolla & Scharf, 2016*) and autocorrelated Kernel Density Estimators (*Calabrese, Fleming & Gurarie, 2016*; *Fleming & Calabrese, 2020*). Data used is from *Marshall et al. (2020)* and can be found in File S4.

duration was too short to confidently identify range stability, which is a prerequisite for beyond sample home range estimates. To help unify the terminology used in reptile spatial ecology studies we draw attention to definitions from existing literature and reiterate them in Fig. 7.

Ambiguous language further compounded reporting issues. Failing to report estimation methods (or reporting with ambiguous or ill-defined acronyms) and associated smoothing parameters completely undermines computational reproducibility and inter-study comparability. Relatively few studies failed to attempt reporting their tracking frequency; many of the 79 failures to determine minimum or maximum tracking frequency were a direct result of ambiguous language, such as using words with multiple definitions (e.g., "bi-weekly") or using imprecise summaries (e.g., "at least"). We also found semantic ambiguity when describing locations. Studies used a wide range of terms to refer to locations, relocations, fixes, datapoints, etc., yet are selecting contrasting or overlapping definitions for these terms. The key distinction for the definition of location is whether it refers each time researchers documented the animal's spatial position or whether it refers only to a unique spatial position (a movement from the previously recorded location) used by the study animal (often referred to as "relocations"). Standardising and unifying terminology is essential for creating widely useful methods and comparable databases (*Schneider et al., 2019*), when in doubt researchers can explicitly define how they are using a given term in the study.

Answering specific questions requires appropriate protocols and, to draw broad inferences among a single study, those protocols must remain consistent. Between study comparisons also require consistency (or at least clear reporting on inconsistencies). The compound effect of geographical, taxonomical, and methodological biases undermine robust generalisations when ignored. Recent macroecological investigations did not explicitly model key methodological variables which affect MCP and KDE home range area estimates (e.g., tracking duration, number of locations, KDE bandwidth selection) (*Slavenko et al., 2016*; *Todd & Nowakowski, 2021*). While the general patterns described in
such studies (e.g., home range area increasing with body mass) likely remain unchanged, not explicitly accounting for varying tracking regimes and different estimation methods (or variation within single estimation method) may obscure more nuanced patterns or differences in space-use. However, both MCPs and KDEs should be avoided when comparing studies for global meta-analyses because of sensitivity to sampling design; whereas methods such as AKDEs explicitly account for movement data biases (*Noonan et al., 2020*).

Researchers should aim to explore the sensitivity of the estimations to researcher choices (*Signer & Fieberg, 2021*), while ensuring that their method suits their question. Home range estimation is not always the correct tool to answer short-term space-use or movement-related questions. In these cases, methods such as step selection analysis (*Avgar et al., 2016*), state-space (*Patterson et al., 2008*) or hidden Markov models (*McClintock & Michelot, 2018*) are more appropriate to infer animal movement, behaviour, and resource selection from telemetry data (*Hooten et al., 2017*). These methods still benefit from accurate methodological reporting and from researchers adopting Open Science principles, as sampling design similarly impacts which methods can apply to a given dataset (*Quick et al., 2019*).

To facilitate detailed reporting of tracking datasets, we have supplied an example report based on an existing tracking dataset (*Marshall et al., 2020*; Fig. S2). This example aims to provide a bare-bones foundation for transparent reporting of sample size, study duration, number of datapoints, as well as important aspects used to describe the tracking regime: namely plots that describe individual tracking durations (while highlighting deviation from proposed tracking protocols), and distribution of time lags between tracks (as a more complete way of describing a tracking regime and the tracking consistency). We have supplied the code (as an .Rmd file, Fig. S3) and data (as a .csv, Fig. S4) used to generate the report as supplementary material.

## CONCLUSION

The past 20 years have seen a growing number of reptile home range studies and continued reliance on traditional but outdated methods, Kernel Density Estimations (KDEs) and Minimum Convex Polygons (MCPs), for home range and space-use estimations, despite the availability of more appropriate methods. Scientific conventions can be slow to shift, and often require substantial interdisciplinary research to move towards better alternatives (*Smaldino & O'Connor, 2020*). We appeal to researchers focusing on reptiles to engage with appropriate statistical methods and Open Science principles, thus maximising the value of hard-won field data. The best way to facilitate broader engagement is to adopt more transparent practices by sharing and fully reporting collected data. Increasing reproducibility and availability of datasets allows researchers to explore beyond home range estimation. Ultimately, we need to match potential research questions to sampling design and the appropriate statistical analyses, achieving a better understanding of both animal movement behavior and their long-term spatial requirements.

## ACKNOWLEDGEMENTS

We thank Suranaree University of Technology, Institute of Science and Institute of Research and Development for logistic support and facilitating our research. We also thank King Mongkut's University of Technology Thonburi for support and particularly the Petchra Pra Jom Klao Scholarship. Also, we thank Aubrey Alamshah for her patience and support during several intensive reviewing and writing gatherings. We thank Supunnee Potijun Strine for suffering through the loss of her husband during long writing weekends and for providing much-needed sustenance during these times.

### Funding

This work was supported by the Institute of Science and Institute of Research and Development for Logistic Support at Suranaree University of Technology as well as the Petchra Pra Jom Klao Scholarship from King Mongkut's University of Technology Thonburi. The funders had no role in study design, data collection and analysis, decision to publish, or preparation of the manuscript.

### Grant Disclosures

The following grant information was disclosed by the authors:
The Institute of Science and Institute of Research and Development for Logistic Support at Suranaree University of Technology.
King Mongkut's University of Technology Thonburi.

### Competing Interests

The authors declare there are no competing interests.

### Author Contributions

- Matthew Crane, Inês Silva and Colin T. Strine conceived and designed the experiments, performed the experiments, analyzed the data, authored or reviewed drafts of the paper, and approved the final draft.
- Benjamin M. Marshall conceived and designed the experiments, performed the experiments, analyzed the data, prepared figures and/or tables, authored or reviewed drafts of the paper, and approved the final draft.

### Data Availability

The data are available at Zenodo: Crane, Matt S., Silva, Inês M.S., Marshall, Benjamin M., & Strine, Colin T. (2020). Supplementary files for Crane et al., ''Lots of movement, little progress'': R Code, data and figures (Version 2.0) [Data set]. Zenodo. http://doi.org/10.5281/zenodo.4303643.

## Supplemental Information

Supplemental information for this article can be found online at http://dx.doi.org/10.7717/peerj.11742#supplemental-information.

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
