# Peer review of "Lots of movement, little progress: a review of reptile home range literature"

_PeerJ, doi:10.7717/peerj.11742_

## Round 0.1 · original submission · Major Revisions

This review provides an overview of home range analysis in reptiles and considers how the methods have advanced over time. I thought this was a worthwhile review that synthesizes an important topic, which has not received much attention recently. The review is well written and comprehensive and has some excellent figures to help summarise the main findings. However, I felt that more could be made out of the data that was collated in the review, particularly with respect to the discussion - which focuses more on how things should be done, rather than discussing why we haven't seen as much movement in these methodological advances as might be desirable, and how this might be overcome.

Overall, the reviewers found strong merit in your review, and had generally positive feedback on your review and how it was written. However, the reviewers have provided some constructive feedback that should help broaden the impact of your article and make the most out of the data you have collated. I encourage you to consider these comments in revising your manuscript.

Reviewer 1 ·

Basic reporting

This manuscript is a review of the home range literature for terrestrial/semi-terrestrial reptiles. Some systematic review methods were used, although the authors refer to this as “a scoping review. Not a systematic review”. The focus is on the characteristics of the studies (e.g. design, taxa, location, methods, etc), rather than home range size itself. There have been previous macro-ecological reviews/studies of lizard, snake and turtle home range size, but I am not aware of anything that has specifically reviewed reptile home range studies from this methodological viewpoint. I believe this review will be of interest to researchers working in the movement ecology field more broadly, not just those working on reptiles.

L123, ideally this terrestriality caveat should be mentioned earlier, including in the abstract. Up until now you have been talking about reptiles in a very general sense, but the focus of this review is on terrestrial/semi-terrestrial species. Is there a reason why you excluded marine species? It would be good to explain that here.
L126–127, be more specific here. Were you only interested in areal, not linear, measures of space use?
L165, collected = recorded
L166-167, I don’t understand what this means? Why is something like mean daily displacement relevant if you are only interested in ‘home range’ estimates.
L170, ‘home range estimation’. Do you mean ‘estimated home range size’?
Table 1, claim = report ?
L182, don’t use the term ‘systematic review’ if it this manuscript is not a SR.
L222, it would be useful to include % in parentheses here.

Experimental design

The design and methods mostly seem appropriate. The authors searched for studies in Web of Science, Google Scholar and Scopus, so there should be good coverage of the peer-reviewed literature. In general the objectives and methods have been explained well.
When I downloaded the dataset I saw that three different people were responsible for extracting the data from the papers, but I couldn’t see where this was explained in the main text. Please add this information, along with an explanation of how you ensure consistent and unbiased data extraction across reviewers.

I was also wondering why did the authors not conduct any statistical analyses to confirm the geographic and taxonomic biases they identified. For instance, chi-squared analyses or some other frequency based method could be used to say compare the number of reptile species present in each country with the number of tracking studies from that country.

Also, by looking at the genera in Fig. S6, it seems that there is a body size bias in the available data (larger genera have more studies). It could be useful to conduct an analysis to quantify this, either at the species or genus level. This body size bias is important because it relates to another comment of mine about how the entire study is framed (see below).

Fig. S1, this is not a proper PRISMA diagram. There should be an extra box under screening that says how many studies were screened and how many were excluded based on the screening. This would explain the 600+ studies that have gone missing between the 2nd and 3rd boxes. The 3rd box on full text eligibility should be under Eligibility, not Screening. http://prisma-statement.org/prismastatement/flowdiagram#:~:text=The%20flow%20diagram%20depicts%20the,and%20the%20reasons%20for%20exclusions.&text=For%20more%20information%20about%20citing%20and%20using%20PRISMA%20click%20here.
L121, please explain why you limited the search to 2000-2019.

Validity of the findings

The whole manuscript, from the title to the discussion, is framed around the idea that there has been little progress in reptile home range studies. I can see how the authors came to this conclusion, but I don’t think it represents a particularly nuanced view of the field. The idea that there has been little progress rests on the fact that many studies still employ MCP or KDE methods for home range estimation. It is true that there are limitations to these approaches and that more advanced techniques have since been developed. But the manuscript does not acknowledge a major reason why researchers still use these techniques: they simply cannot obtain enough data to use more advanced, data-hungry methods such as Brownian bridge movement models and hidden markov movement models. Most reptiles are small, too small to carry GPS trackers. This means that the volume of tracking data that can be collected is constrained when using VHF tracking. Although there are some very small GPS trackers, e.g. <5 g or <1 g, the battery life is very short and they are very expensive.

I think the authors should add some analyses to quantify these additional biases and add a paragraph to the discussion covering these issues. I thought you might be able to compare the weights of species for studies that used traditional methods vs. more advanced methods (e.g. BBMM), but I could only see five of the latter studies in the data. They all seem to be non-small species, so perhaps you could incorporate this into the new discussion paragraph I suggested. Another way to quantify this bias would be to look at tracking durations and data volumes as a function of body size. I would expect that larger species are tracked for longer and have more data available per individual.

Additionally, sampling a broader time period, say beginning in 1990, would enable the authors to provide a better judgement of changes in the field. Inspecting Fig. 2b shows that the proportion of studies only using MCP has decreased over time, so arguably this is a form of progress. You could actually add a statistical analysis to quantify this change. The adoption of new techniques is often a slow process, so it’s not really surprising that there isn’t an influx of studies using Brownian bridge movement models yet, particularly for reptiles, which are already understudied.

L106 says “to assess whether the field showed a similar trajectory of increasing data volume”, but no temporal analysis of changes in data volumes is provided. This would be a very valuable addition and is related to my points above.

Fig. 2, it is not surprising that the number of studies increased over time. The number of studies in Scopus using (ecology OR conservation) increased from 12,500 in 2000 to 60,000 in 2019. This actually suggests that reptile tracking studies are a growing at a slower rate than ecology/conservation in general. You could add this information as an extra y-axis with a line showing growth in the broader literature over time.

Additional comments

L17, “Reptiles are the most species-rich vertebrate group”. Do you mean terrestrial vertebrate group? There are more species of fish than reptiles.
L46-48, these numbers from Bohm et al. 2013 are a bit outdated now given the number of species assessed for the Red List has increased substantially in the past seven years. Using the advanced search function on the RL website shows that 15% are classed as DD and 18% threatened (CR, EN, VU).
L55, are the efforts or the researchers frustrated? Need to use a different word here.
L56, “during visual count surveys”. Other survey techniques also suffer from these issues (e.g. pitfall trapping).
L61, it would be good to add an extra sentence or two here rounding out the logic that this information can aid conservation efforts. The previous sentences have talked about reptile population declines and lack of data hampering conservation. I would add here some concrete examples where telemetry data has actually informed reptile conservation.
L65, ‘analytical method complexity’. This is confusing. Suggest spelling it out more clearly, perhaps along the lines of ‘broaden the range of analytical tools that can be used’ (if this is what you mean).
L66 and 67, ‘movement-based methods’. What does this mean?
L85, lead should be led
L101, there are more than 1000 reptile species in Australia. Besides, to illustrate the bias it makes more sense to report the 39% figure for reptiles from Goldingay (cf. 16% for mammals).
L104, I don’t think these figures illustrate a lack of advancement. If the % using KDE was 45% in 2001–12 (Goldingay), but 20% in say an earlier period of 1980–1995, then arguably there has been advancement.
Fig. 3, the colour scale for the main map is missing.
Fig. 3, please explain which groups were used to create the reptile diversity map. Was it all species except sea turtles and sea snakes?
L311, ‘being’. Is this the right term? I can’t work out the meaning here.
L316, the wording of this sentence is confusing. Suggest rewriting it.
L339, how does zero-inflation affect tracking datasets? Tracking data in its simplest form contains coordinates, date and time, none of which can be zero. Please be clearer here as to whether you are talking about derived datasets or something else.
L348, ‘significance’. Relevance, accuracy or representativeness may be better terms.
L390. ‘have failed to account for’. I strongly suggest using less inflammatory language here. How about ‘have not accounted for’.

Reviewer 2 ·

Basic reporting

• Clear, unambiguous, professional English is used throughout and easy to understand.
• References are included where appropriate with sufficient detail.
• Article structure with figures and tables follows conventional patterns. Raw data is shared via a link.
• The review is of interest, adds to the body of knowledge and within the scope of the journal.
• This review is unique in scope and does not repeat recently published meta analyses of this type.
• Introduction is robust and outlines justification for this research (see additional comments below).

Experimental design

• Analysis methodology is well detailed and presented in an easy to understand format.
• Sources are cited well with sufficient detail in the placement of references.
• Paragraphs in sections flow well into one another based on topic.

Validity of the findings

• Analysis conclusions mirror the justification set forth in the Introduction.
• Future recommendations are in sufficient detail with external links provided. An expansion of data reporting guidelines is needed.

Additional comments

Review Summary:
• Clear, unambiguous, professional English is used throughout and easy to understand.
• References are included where appropriate with sufficient detail.
• Article structure with figures and tables follows conventional patterns. Raw data is shared via a link.
• The review is of interest, adds to the body of knowledge and within the scope of the journal.
• This review is unique in scope and does not repeat recently published meta analyses of this type.
• Introduction is robust and outlines justification for this research (see additional comments below).
• Analysis methodology is well detailed and presented in an easy to understand format.
• Sources are cited well with sufficient detail in the placement of references.
• Paragraphs in sections flow well into one another based on topic.
• Analysis conclusions mirror the justification set forth in the Introduction.
• Future recommendations are in sufficient detail with external links provided.

Overall summary: Well composed and detailed with sufficient justification included. The authors did very well in explaining their study parameters regarding the inclusion of articles for this meta analysis to highlight lacking data for reptile species. One suggestion for improvement is to expand upon the “Example Report” as a guideline for future studies involving telemetry of reptile species.

Detailed feedback:
Line 45: Include examples of threats.
Line 50: Expand here on the relationship between collapses in diversity and data needs for effective understanding of spatial ecology.
Lines 50 – 54: Reference formatting inconsistent.
Line 67: Define what you mean by “movement based methods” here for the reader to lead into the following paragraphs.
Line 103: Use MCPs instead of ‘the MCPs’ throughout.
Line 106: A paragraph here on the geographic and/or taxonomic biases in telemetry studies of reptiles and others will help to make your justification for this research more impactful.
Line 123: Great explanation of your methodology and decisions.
Line 179: Sentences are redundant and lacking end parentheses.
Lines 180-282: Well written and composed with sufficient detail included.
Line 295: Conclusions are supported by data and external articles.
Line 308: Reference formatting inconsistent.
Line 316: A mention of the need for data sharing should be mentioned at the end of the introduction.
Line 318: A statement here comparing availability of reptile data vs. ornithological or mammalian data on MoveBank would support this statement.
Line 398: More details on recommended data reporting would be helpful here.
Line 393: Supplemental File 2 is not included in my materials.
Line 468: Should be Averill-Murray RC, Fleming… Check reference list for formatting consistency.

Reviewer 3 ·

Basic reporting

• Clear and unambiguous professional English throughout
o yes

• Lit references/sufficient background/context
o Light on movement literature. Why for instance, are some of the foundational papers on space use cited? I’m thinking of Mevin Hooten’s book, and several refs therein—from Moorcroft’s work, to Geert Aarts’ work, to Devin Johnson’s work, to Ephraim Hanks work on continuous time Markov chains. There are a lot of papers in the statistical ecology literature that talk about other approaches and I feel these could be included here.

• Prof. structure, figures, tables, etc. Raw data shared
o Line 398 indicates the code is shared, but I didn’t see it in the supplemental folder I had access too. Nevertheless, I applaud the authors for making this call and encouraging users to produce their work in more reproducible fashion.

• Is the review of broad and cross-disc interest and within the scope of the journal?
o To me it felt primarily targeted towards herpetologists. I was aware of the issues raised w/r/t the methodological approaches and since I don’t study this taxa, I don’t have a good feel for the reach outside of this audience.

• Has the field been reviewed recently
o Seems like from the papers cited in the intro that one is due

• Does the intro adequately introduce the subject and make it clear who the audience is and what the motivation is?
o Pretty good, but for me got better on second reading. Motivation could have been clearer. How does this apply to conservation, which is lacking.

Experimental design

• Article content within Aims and Scope of journal
o yes

• Rigorous investigation
o Yes, thorough in terms of the steps taken, terms used, etc.

• Methods described with sufficient detail
o Assume yes. Code is said to be supplied.

• Survey methods consistent and comprehensive?
o Not entirely on the movement side. Lit search was unbiased. I liked how the authors highlight geographic and taxonomic biases. Figure 3 is really helpful.

• Sources adequately cited?
o For the most part, save for movement lit

• Logical organization?
o Yes

Validity of the findings

• Impact and novelty not assessed
o No comment - wasn’t really sure how to answer this question.

• Conclusions well stated, linked to questions, and limited to supporting results
o Well stated in regards questions, but questions needed improvement/expansion

• Speculation?
o No comment

• Does the argument meet the goals set out in the Intro?
o Yes, but goals could have been stronger and broader

• Does conclusion identify unresolved questions/gaps/future directions?
o Could be stronger.

Additional comments

Crane et al. review the literature on the estimation of home ranges in reptiles, and based on their search find that: 1) the study of home ranges are biased across geographic and taxonomic axes; 2) users who do estimate home ranges are often using outdated methodological approaches; and 3) that users are failing to provide adequate documentation about their methods—both for data collection and for statistical analyses—to facilitate broader meta-analytic approaches. Crane et al. conclude by urging users to use more modern approaches and to do so in ways consistent with reproducible research. The study brings up interesting points that reptiles are much less represented in spatial studies than mammals and birds, and it highlights the bias towards temperate western hemisphere species. The authors should expand on why it’s important to have more studies on tropical reptiles - for instance because the tropics are experiencing more rapid loss of biodiversity. Since there are so many more studies on mammals, this paper should mention how aspects of mammal spatial ecology could be applied to reptiles. Overall, I liked this paper and feel that it’s worthy of publication, but not before some serious revisions are made.

General Comments
I like this paper, and I feel that the authors have done a nice job assembling it and presenting their findings. I also think that it could be made stronger, and appeal to a broader audience with some major revision. In particular my issues are mostly along the lines of the presentation of the issue(s) raised by the authors. The introduction, to me, reads a bit scattered, so it’s hard to get a good feeling, specifically, why a review is needed and what, as a reader, we can expect from the review. Several questions came to mind when I was reading the intro: why do we want to know about the spatial ecology of reptiles? How would knowing more about the movement, and the summary of movement data, help reptiles? Assuming we do want to know, why do we only focus on home ranges? (It’s important to recognize that different movement features happen at different scales, and within the home range, there may be context specific areas and behaviors that are critical for life history. So it would help orient the reader to why this particular spatial summary was chosen.) What are we trying to find and then show in the review. The authors state this on 106-08, but to me anyway, haven’t made the case why these questions need to be answered with a new review. I think it the authors better make the case for a) why the spatial ecology is important, and b) why existing study of it is lacking, then it will be more obvious why the review is needed. Further, it will be clearer when recommendations are stated later in the ms, how they resolve the issues proposed in the intro.

I don’t study reptiles, so have no feel for the diversity/geography tradeoff, nor the allocation of studies across different taxa. This may be well known for readers within the reptile audience, but for readers outside that, it’s not clear why this is a major issue.

Similarly, for the non-quantitative practitioner, some more clarification on the statistics might help the reader. It would help those readers without a strong GIS or statistics background to understand aspects of the study like Figure 2, which was otherwise an effective visualization. However, it was difficult to know what study procedures were better (VHF vs GPS, MCP vs KDE) and why.

Specific Comments
Line 17 of the abstract is incorrect - fish are the most species-rich vertebrate group, not reptiles.

Line 60 – what is meant by “valuable insight?” This part of the intro could be clearer about the specifics of what a) telemetry, and b) proper analysis of telemetry will help tell about reptiles, their ecology, and their conservation. In addition, it’s possible to make the intro appeal to a broader audience here by highlight some other systems where this has been done well.

Line 67 – stagnated, really? It seems like a pretty vibrant field to me, so I was surprised to read this. Do you mean that the uptake of modern techniques by practitioners is stagnating? If so, state that, otherwise I think this is a little misleading. This is kind of what you are implying in 89-90.

Line 92 – if this is the last big review in the field, then I think the authors should stress what new we are to learn with this new review herein.

Line 112 – nice to see this stressed here and throughout the ms – the authors are to be congratulated.

Line 198-99 – it’s nice to see the discussion of the frequency and programming of the tags, as this can be an important question. There’s often quite a tradeoff in the questions you can ask vs. the data you can collect. Some discussion of this (see for example Quick et al. 2019. Mind the gap—optimizing satellite tag settings for time series analysis of foraging dives in Cuvier’s beaked whales ( Ziphius cavirostris ). Animal Biotelemetry 7:5) would help put this issue in context. Further, and importantly from the practitioner point of view, a lot of the Hidden Markov Model machinery can handle these issues (in discrete time anyway). Relatively straightforward model-fitting approaches are attainable, and should be cited, i.e.

McClintock, B. T., and T. Michelot. 2018. momentuHMM: R package for generalized hidden Markov models of animal movement. Methods in ecology and evolution / British Ecological Society 9:1518–1530.

Michelot, T., R. Langrock, and T. A. Patterson. 2016. moveHMM: an R package for the statistical modelling of animal movement data using hidden Markov models. Methods in ecology and evolution / British Ecological Society 7:1308–1315.

While still on the data collection discussion, the authors should also take into consideration cost while recommending higher sample sizes and study durations.

Figure 4 and the accompanying data in the “Geographic and taxonomic biases” section starting at line 257 is interesting, but some of it is an oversimplification. The authors seem to be suggesting that the percentage of species studied of each clade should be proportional to the size of the clade. This overlooks advantages of thoroughly studying and understanding a single clade, like Crocodylia. Furthermore, it would benefit from having much more specific conclusions on how improving reptile spatial studies aids in conservation. This would help tie in Figure 5 more and show how these two space-use estimates directly apply to reptile conservation.

I particularly liked Figure S1 which helped me visualize how the studies were chosen and filtered. However, I felt that Figure S2, the word cloud, didn’t add much.

The analysis of terms used in studies (line 377 to 380) seemed overly nitpicky, detracting focus away from quantifiable aspects of the data reporting. I like the author’s call on 383 for standardization

---

## Round 0.2 · Minor Revisions

Both myself and the reviewers are satisfied with the changes that have been made to the manuscript - which have strengthened the manuscript.

One of the reviewers has identified a few further changes for you to address - please consider these as you prepare a revised version of the manuscript

Reviewer 1 ·

Basic reporting

No comment.

Experimental design

No comment.

Validity of the findings

No comment.

Additional comments

The authors have done a good job of responding to the earlier reviewer comments, including mine. Many of the arguments are much better supported now the thinking has been fleshed out more. The figures in this paper are top notch, well done! I just have some relatively minor comments, detailed below.

L12, methodology for *studying* reptiles ?
L44, that Tucker reference is good, but you might consider supplementing or replacing it with something that has data on reptiles. e.g.:
https://academic.oup.com/jue/article/6/1/juaa014/5867359?login=true
https://www.nature.com/articles/s41559-020-01380-1
https://onlinelibrary.wiley.com/doi/epdf/10.1111/geb.13225
L47, nature = natural
L223, "Such reporting is key when measurements of movement capacity are calculated". It's not clear what this means. Maybe save it for the Discussion anyway.
L274, I suggest rewriting this sentence to change the order. It is confusing to say all other countries were low, but then reveal that x y z broke the trend. "All other countries are dramatically lower (<8 studies, 30 countries with a single study), with only Australia (35), Canada (22), and South Africa (12) breaking the trend."
L300, 'at a population level'. I think you have adopted this term from brms, but it is confusing to use it here in an ecology paper. Suggest changing/removing it.
L308, 'decisions on' should be 'decisions about' or 'decisions regarding'
L325, somewhere here it could be worth mentioning that there is a negative relationship between number of genera within a major clade and proportion studied (L289 in the results). In some ways it is not surprising that only a small number of the 500+ snake and lizard genera have been studied.
L349, this is also relevant here https://www.sciencedirect.com/science/article/abs/pii/S0169534718302817
Fig S1: the box ‘Reports sought for retrieval’ should have 413, not 1028.
Supp File 2, is there a reason why the species name is not included? It could be worth adding a little preamble to this explaining why the file is included (so that it can stand on its own independent of the main text). Further, as per your own recommendation, I suggest you define your terms in this document. It is confusing to say at the start you located animals every 8 or so days, but in figure 2 you refer to this as ‘Time lag between tracks’. Figure 1, I can only see the different black and red lines if I zoom in 1000%. Suggest you make that figure fill a whole page, or choose a different method to show the differences.

Reviewer 3 ·

Basic reporting

no comment

Experimental design

no comment

Validity of the findings

no comment

Additional comments

The authors are to be commended for making a great effort to respond to all my comments as well as the those of the other reviewers. The manuscript is much stronger now, and I recommend it for publication.

---

## Round 0.3 · accepted · Accept

Thank you for making the requested changes so quickly - I am now happy with the manuscript as it is and pleased to be able to accept it for publication. I have noted your request for shared first authorship, and have passed that on to the editing staff for action.